Disjoint combinations profiling (DCP): a new method for the prediction of antibody CDR conformation from sequence

Nikoloudis Dimitris 1
Pitts Jim E. 1
Saldanha José W. 2 jsaldan@nimr.mrc.ac.uk
1 Department of Biological Sciences, Birkbeck College, University of London , London , UK
2 Division of Mathematical Biology, National Institute for Medical Research , London , UK
Martens Lennart
Electronic publication date: 2014 Jul 1
Publication date: 2014
Volume: 2
Electronic Location ID: e455
Received 2014 Mar 26; Accepted 2014 Jun 5
Copyright: © 2014 Nikoloudis et al.
Copyright year: 2014
Copyright holder: Nikoloudis et al.
License: This is an open access article distributed under the terms of the Creative Commons Attribution License, which permits unrestricted use, distribution, reproduction and adaptation in any medium and for any purpose provided that it is properly attributed. For attribution, the original author(s), title, publication source (PeerJ) and either DOI or URL of the article must be cited.
License URL: https://creativecommons.org/licenses/by/4.0/

Keywords: Conformational prediction, CDR conformation, Blind test, Canonical templates, CDR-H3 sequence rules, DCP, Humanisation, Prediction from sequence, Antibody engineering

Funding: The authors declare no external funding sources.

==============================
The accurate prediction of the conformation of Complementarity-Determining Regions (CDRs) is important in modelling antibodies for protein engineering applications. Specifically, the Canonical paradigm has proved successful in predicting the CDR conformation in antibody variable regions. It relies on canonical templates which detail allowed residues at key positions in the variable region framework or in the CDR itself for 5 of the 6 CDRs. While no templates have as yet been defined for the hypervariable CDR-H3, instead, reliable sequence rules have been devised for predicting the base of the CDR-H3 loop. Here a new method termed Disjoint Combinations Profiling (DCP) is presented, which contributes a considerable advance in the prediction of CDR conformations. This novel method is explained and compared with canonical templates and sequence rules in a 3-way blind prediction. DCP achieved 93% accuracy over 951 blind predictions and showed an improvement in cumulative accuracy compared to predictions with canonical templates or sequence rules. In addition to its overall improvement in prediction accuracy, it is suggested that DCP is open to better implementations in the future and that it can improve as more antibody structures are deposited in the databank. In contrast, it is argued that canonical templates and sequence rules may have reached their peak.

Introduction

Antibodies can recognise virtually any given molecule mainly by variation in the length and sequence of their Complementarity-Determining Regions (CDRs), which form the antibody’s binding interface. Three CDRs are found in the antibody’s Heavy chain (CDR-H1, -H2, -H3) and three in the Light chain (CDR-L1, -L2, -L3). The first definition of CDRs was by Wu & Kabat (1970) while performing an analysis of the variable domains of Bence-Jones proteins and myeloma Light chains. Later, Kabat and colleagues compared the sequences of the hypervariable regions in the then known structures and observed that at 13 sites in the Light and 7 in the Heavy chains (Kabat, Wu & Bilofsky, 1977), the residues are conserved. They suggested that these positions in the sequence are involved with structure rather than specificity, introducing for the first time a possible relationship between sequence and loop conformation in antibodies. A second set of observations of the crystal structures of Fab fragments and myeloma proteins revealed that, in many cases, hypervariable regions with the same length but different sequences have the same main chain conformation (de la Paz et al., 1986).

It was in 1986 (Chothia et al., 1986) that specific residues were directly associated with the conformation of the hypervariable regions during a visual analysis of the sequence and structure of antibody D1.3, thus introducing the notion of the “canonical model”. From this point, various further studies enriched the table of structurally-determining residues (canonical residues), by observing the amino acid similarities at key interacting positions within sequences of members of any given conformational class, of the known and newly defined canonical structures, for the three CDRs in Light and the first two in Heavy chains (Chothia & Lesk, 1987; Chothia et al., 1989; Chothia et al., 1992; Barré et al., 1994; Tomlinson et al., 1995; Guarne et al., 1996; Martin & Thornton, 1996; Morea, Lesk & Tramontano, 2000; Vargas-Madrazo & Paz-García, 2002). Therefore, these collections of structurally-determining residues created canonical templates for each known conformational class, which defined the allowed residues per identified position in the variable chain. These canonical templates could then be used for prediction, from sequence alone, of the conformation of a new CDR by requiring its variable chains match as many, if not all, of the allowed residues present in the template. Regarding the sixth and final CDR-H3, a number of studies (Shirai, Kidera & Nakamura, 1996; Shirai, Kidera & Nakamura, 1999; Furukawa et al., 2001; Kuroda et al., 2008) provided structure-determining sequence rules for the prediction of the CDR-H3-base (or ‘take-off’, ‘torso’ or ‘anchor’) conformation.

In the latest relevant study (North, Lehmann & Dunbrack, 2011), it was inferred that the effect of canonical residue overlap between templates caused by the proliferation of structures was diminishing the efficacy of the canonical model. Instead, a mixed approach was proposed for prediction of CDR conformation, sometimes based on the presence of a very small number of statistically prominent structurally-determining residues, the gene source, CDR length or even the use of Hidden Markov Models (HMMs). Therefore most conformational clusters/classes were noted as not canonical, while a considerable number were characterised as non-predictable altogether. Furthermore, concerns were raised regarding the predictability from sequence of the bulged (including double-bulged) CDR-H3-base conformation.

The accurate prediction of CDR conformation is important in modelling antibodies for protein engineering applications (e.g., ab initio design of antibodies, antibody humanisation, vaccine design, etc.). Specifically, knowledge of the CDR conformation is crucial for the creation of a stable binding interface, modification of the antibody’s binding affinity or even identification of an epitope. Computational methods such as the canonical model or CDR-H3 sequence rules, which attempt conformational prediction of CDRs from sequence alone, have the advantage of being inexpensive and fast while requiring only a simple input; their major drawback being the inability to predict conformations that were never observed before experimentally. In this context, a re-evaluation of the performance of the canonical model in predicting the class of CDR conformation from sequence alone is presented in light of the latest new and multi-level complete CDR clustering (Nikoloudis, Pitts & Saldanha, 2014). The key residues are updated in the existing canonical templates from the sequences of members of each level-1 cluster/class, and correspondingly the canonical templates for new clusters in a given length are populated, using the key positions defined for that length by Martin & Thornton (1996). Those defined key positions are identical for all clusters of a given length. In this way, an assessment as to whether the canonical model is still effective as the quickest and simplest prediction method for antibody CDR conformation is carried out, and the effect of canonical residues’ overlap between templates caused by the proliferation of cluster sequence populations can be evaluated.

For the hypervariable (both in sequence and conformation) CDR-H3, the sequence rules for CDR-H3-base prediction described in Shirai, Kidera & Nakamura (1999) are tested, as well as their updated versions in Kuroda et al. (2008). The goal here is to compare the accuracy of the two sets of rules and, more importantly, to find out if the continual adaptation to new sequences with additional rules, exceptions and overrides is beneficial to this predictive model.

Besides testing these two popular and historic approaches on an updated dataset, a new predictive model from sequence alone is also introduced which aims to bring improved accuracy over previous sequence-based methods, while retaining their rapid execution and simplicity of usage. All the characteristics of the new method are detailed, step-by-step: inception, goals, basic concepts and definitions, implementation strategies, training and prediction workflows. A demonstration is presented of a standard predictive model derived from the method as well as an assessment of its efficacy on the same set of CDRs employed for the testing of the canonical model and CDR-H3-base rules. As this new method allows parameterisation, future dedicated work could take advantage of the general framework provided and propose a number of different or improved implementations.

The prediction results obtained by the new method are directly compared to those from previous approaches and complemented by statistical characteristics of the training, validation and test sets. Additionally, special importance is attributed to each method’s performance in predicting the major cluster/conformation (class-I) in any given CDR/length combination (e.g., CDR-L1 11-residues). Indeed, as is revealed by the population percentages per cluster in Nikoloudis, Pitts & Saldanha (2014), in each CDR/length with more than 10 unique sequences there is usually a single cluster which regroups the large majority of the known conformations, while the remaining fraction may be populating a considerable number of much smaller clusters. In the 15 lengths (first 5 CDRs) that contained more than 10 unique sequences in their clustered population and produced more than one cluster, the major cluster of each length represented on average 74% of the available unique sequences (median: 86%). As a consequence, these major conformations are expected to occur more frequently and are accordingly more probable to prove of interest in research scenarios. For this reason further analysis is undertaken of the prediction results to calculate the precision, recall and F-measure for all major clusters, and the corresponding comparisons between methods are presented.

Methods

A new blind dataset

As the clustering dataset in Nikoloudis, Pitts & Saldanha (2014) was locked on the 31/12/2011 edition of the PDB (Berman et al., 2000), this presented an opportunity to conduct a true blind-testing by downloading the antibody structures that were released subsequently. Hence for the new dataset, a search was performed in the PDB for structures released between 01/01/2012 and 21/11/2013, using the same methodology as in Nikoloudis, Pitts & Saldanha (2014), which returned 312 files, two of which contained structures from 3 antibodies (PDB codes 3ULU, 3ULV). After removing redundant sequences, there remained a total of 230 antibody structures: 210 had both Heavy and Light chains, 4 had only a Light chain and 16 had only a Heavy chain. All redundant instances (i.e., multiple copies of the same CDR sequence within the same structure and CDRs from different structure files with identical Fv sequences) were additionally searched for different CDR conformations. Only one of the 230 structures was retained despite the fact that it was redundant (4DN4), because a different CDR-L1 conformation was observed between the two crystal structures (4DN3/4DN4, free and bound versions, respectively).

As DCP required parameter tuning, a validation step had to be inserted. However, since the initial structure of the data to be predicted presented a majority of clusters with only between one to three unique sequences, it proved impractical to perform a traditional k-fold cross-validation on the clustered set as these smaller clusters could not be further subdivided in a meaningful way. Instead a 3-way experiment was designed, where the previously clustered dataset was used for training, while the new dataset was divided approximately in half into a validation set and a test set. The validation set comprised of all PDB files released between 01/01/2012 and 14/03/2013 (113 non-redundant antibody structures), while the test set included all the subsequently released structures (15/03/2013–21/11/2013, 117 non-redundant antibody structures). This division of the dataset by time preserved the double-blind nature of the experiments, since the complete test dataset was also constructed with time of release as the sole criterion, thus eliminating any subjectivity from the selection, analysis and interpretation.

An examination of the redundant sequence content (complete Fv identity) between training and test datasets was also performed. This count revealed a 7%–9% fully redundant sequence content in the test dataset (i.e., present in the training dataset) in all considered CDRs (specifically, full count[subset-1 count/subset-2 count]: 11[4/7], 13[7/6], 17[7/10], 17[7/10] and 17[7/10], for CDR-L1, CDR-L3, CDR-H1, CDR-H2, CDR-H3-base, respectively, Supplemental Information 3). While the fully redundant content appeared to be relatively low, the concerned entries were retained in the test dataset in order to allow an appreciation of the methods’ accuracy in predicting a trained sequence and demonstrate their capacity to overcome overlapping predictive definitions.

By using this new dataset, it was possible to retain the previous entire clustered set as a prior knowledgebase and to assess the sequence-based prediction methods in realistic conditions without discarding or ignoring any data, both during training/updating and testing. This ensured that DCP training and canonical templates’ updating remained blind toward the new PDB files. In terms of predictions with canonical templates, the entire new dataset served for testing since no validation step was required. However, for practical reasons, the above first subset will henceforth be called “the validation set” (for DCP) and the second subset “the test set” (for DCP), despite the fact that both constitute test sets for the canonical method.

All conformational predictions were applied at the first level of the clustering set’s nested scheme. New Fv sequences were numbered, using the numbering scheme and CDR extents described in Nikoloudis, Pitts & Saldanha (2014). The Cα-backbones of new CDRs were then successively superposed onto the medoid structure of every cluster of the same length, in order to determine the actual conformation of new CDRs. For a new CDR to be assigned to a pre-existing conformational cluster, its RMSD to the cluster’s medoid was required to be lower than the cluster’s radius.

A new method for prediction of CDR conformation from sequence

Method presentation

It has been made clear through various studies (Chothia et al., 1989; Chothia et al., 1992; Alzari et al., 1990; Al-Lazikani, Lesk & Chothia, 1997; Martin & Thornton, 1996; Morea, Lesk & Tramontano, 2000; Vargas-Madrazo & Paz-García, 2002; Shirai, Kidera & Nakamura, 1996; Shirai, Kidera & Nakamura, 1999; Kuroda et al., 2008) that the CDR sequence is not always solely determinant of the CDR conformation. Several residues external to the CDR, from the framework, other CDRs or the second Fv chain, were retained as structurally-determinant and included in predictive canonical templates or sequence rules. These residues were spotted after pedantic visual examination of a number of antibody structures of interest, as making important contacts with CDR residues. However, this process can potentially lead to misleading generalisations due to crystal errors, or the intrinsic backbone and side-chain flexibility of surface residues such as those in CDR sequences.

In the new method now presented, a generalisation for the presence of class-specific combinations of residues is proposed. These combinations of residues would represent conformation-influencing synergies that are expected to appear exclusively or preferentially in members of one cluster. As far as the physico-chemical aspect of the residues’ interaction is concerned, these combinations may be representing steric effects, creation of a hydrophobic pocket or local environment, hydrogen-bonding, van der Waals’ contacts, salt bridges, backbone flexibilities, etc. Of course any investigation of sequence sets with such physico-chemical criteria would dramatically increase the complexity of any method. Instead a simpler model is proposed where the nature of these interactions, as well as the very residues which participate, remain irrelevant to the prediction procedure. More specifically, it would be of interest to search for those combinations of positions in the antibody Fv sequences that contain combinations of residues that are always different between different conformational clusters, i.e., combinations of positions that present disjoint combinations of residues between classes. In this way the sequence differences between different classes are examined, instead of the sequence similarities within a class as is the case with the canonical model. This approach was named ‘Disjoint Combinations Profiling’, or DCP, and all its characteristics are further detailed in the following sections.

Basic definitions

For the formulation of this new method a number of novel features needed to be defined, which are detailed later. The basic terms used in the DCP prediction method are provided here in Table 1, as both an introduction and for quick reference.

Table 1 DCP terms and definitions.

A list of terms that were used for the formulation of the DCP method and their definitions.

Interaction Frame (IF)	A list of Fv positions that are found in the neighbourhood of the
examined CDR, its residues included	
IF sequence	A sequence of residues derived from an antibody’s Fv that
correspond to the positions included in the IF	
Query IF sequence set	A group of non-redundant IF sequences from all members of
the cluster that is being profiled	
Target IF sequence set	A group of non-redundant IF sequences from the members of all
clusters in the examined length, excluding the cluster that is being profiled	
IF fragment	A singlet or a non-necessarily consecutive combination of IF positions
(couplet/triplet/quadruplet/etc.)	
IF fragment sequence	The corresponding sequence of residues in an IF fragment	
Query/Target
fragment sequence	IF fragment sequences from the Query/Target IF sequence
sets form Query/Target fragment sequences, respectively	
Signature signal	An IF fragment that presents disjoint IF fragment sequences
between Query and Target sets	
DCP signature	The complete set of signature signals that are consequently
used for the prediction of a given cluster	

DCP setting-up and training

In this demonstration of DCP, all neighbouring residues of a CDR are included, within a radius of 4 Å, 6 Å or 8 Å, as potentially interacting with the CDR in a way that is influencing its conformation. The initial assumption is that these neighbourhoods of members of the same conformational cluster have equivalent influence on the observed conformation. Therefore, it is expected that within these neighbourhoods there exist combinations of positions that make distinct conformational-influencing synergies, and whose sequences are never observed in members of a different cluster. These synergies could be caused by any number of the aforementioned residue-to-residue interactions. The theoretical basis behind this parameter could be the chained influence that residues may have on a local conformational feature, also implicating residues that make indirect contact with the CDR; e.g., a cascade of interactions between 3 or 4 residues where the last residue resides on the CDR but makes no contact whatsoever with the first residue of the cascade. It is therefore possible that DCP captures such chained synergies, which are different between different conformational classes.

All the Fv positions that are predominantly found within the selected radius of an examined CDR, its residues included, define its ‘Interaction Frame’ (IF). This frame of positions was constructed after visual examination with the graphics program Swiss-PdbViewer (Spdbv; Guex & Peitsch, 1997) of a large number of antibody structures. During visual examination, all positions that satisfied the radius criterion and were common to all members of all clusters, were retained. As the antibody framework is very stable, the vast majority of neighbouring positions that were observed (over 90%) was topologically preserved between the examined CDRs. This operation was repeated for each CDR.

Once the IF is selected for a given CDR, the sequences of all cluster members per CDR/length combination are parsed for the residues that occupy the Fv positions found in the IF. These residues are then arranged in the same order as the respective positions appear in the IF, in order to form the corresponding ‘IF sequence’. This way, each cluster now has a set of IF sequences that can be compared with each other for the detection of disjoint combinations of residues between them. A graphical representation of these setting-up steps can be seen in Fig. 1.

Figure 1 Preparatory steps for DCP.

Here, an Interaction Frame (IF) is selected for CDR-L1 and the corresponding IF sequences are synthesised for each one of the four clusters of the given length. For computational reasons the same IF is defined for all lengths of any given CDR (here, CDR-L1 for illustration purposes). Therefore, observed gaps in IF sequences correspond to insertions populated in longer lengths than the one shown in the illustrated example—gaps are filled accordingly in those lengths’ IF sequences. Spare gaps, on the other hand, may correspond to IF positions pointing to unpopulated insertions from other CDRs or deletions in the Fv sequence. Also, gaps are present if there is no Light or Heavy chain in that particular structure. Positions at the end of the IF, marked as ‘n−x’, refer to CDR-H3 positions at a sequential distance x from the last residue n (H102; see text).

A common problem in CDR conformational prediction from sequence alone is the presence of sparsely populated clusters/classes. The sequence examples of those clusters are often so few that it becomes impossible to detect sequence features that are at the same time common between members of that cluster but different from other clusters. Especially so when the major cluster in a given length also has few members; any comparisons between the different clusters’ sequences become prohibitively risky. For the DCP training process, this obstacle was overcome by regrouping the sequences of all clusters in that length, except for the one that is being profiled. Indeed, in searching for differences, the profiled class needs to be presented against an ‘anti-reference’ rather than a traditional ‘reference’ used in many prediction methods. For example, it is possible to screen class A against what “is_not_class_A”, so by regrouping all “non_class_A” instances there is a practical enrichment of the volume of sets of sequences to be compared.

The ‘Query IF sequence set’ was defined as the group of non-redundant IF sequences of all members of the cluster under examination and the ‘Target IF sequence set’ was defined as a group of non-redundant IF sequences from members of all clusters except for the one that is being profiled. For example, when examining cluster-1 in a CDR/length with 4 clusters, a comparison is made of Query IF sequence set [cluster-1] versus Target IF sequence set [clusters-2/3/4]. The profiling for disjoint combinations can then be initiated by cycling through all combinations of Fv positions within the IF, up to the maximum combinatorial order that is pre-selected (e.g., singlets, couplets, triplets, quadruplets, or quintets, etc., each time including combinations of lower order), and extracting the corresponding amino acid sequences from the Query/Target IF sequence sets. Each combination of positions was called an ‘IF fragment’ and, accordingly, the corresponding extracted residues formed an ‘IF fragment sequence’.

Once all respective amino acid fragment sequences are acquired from both Query and Target sets, the corresponding fragment sets are then examined for disjointness, i.e., that no sequence fragment is shared between the two sets. If the sets prove to be disjoint, that IF fragment is retained as pointing to a potentially significant difference between the two sets. This IF fragment is called a ‘Signature signal’. The rationale is that if any sequence combination of the examined IF fragment is shared even once between the members of the different clusters, then the examined IF fragment sequences are not mutually exclusive and therefore cannot be theoretically considered as unique to any conformation. The complete list of signature signals constitutes the ‘DCP signature’ of the examined (Query) cluster/class, which is consequently used for its prediction with new sequences. A graphic representation of this training process can be seen in Fig. 2.

Figure 2 The training procedure using Disjoint Combinations Profiling.

Definition of Query and Target IF sequence sets, extraction of all available IF fragment sequences and comparison between corresponding sets of fragments for disjointness, leading to signature signals.

As a note, the basic properties of combinations imply that the observance of any signature signal of lower order automatically renders equally disjoint any combination of greater order, which contains all the IF positions of the lower order combination. For example, when IF fragment L90–L95 is disjoint, thus becoming a signature signal, any higher order combinations containing the previous IF positions are also disjoint; e.g., L90–L91–L95, L89–L90–L95, L89–L90–L91–L95, etc., are all equally signature signals. Therefore, in order to avoid unnecessary redundancies within a DCP signature which may affect prediction scoring, a filtering is performed that removes signature signals from the DCP signature when they contain other signals of lower order.

Prediction of CDR conformation with DCP signatures

Once a DCP signature and a Target IF sequence set are acquired for each conformational class, it becomes possible to predict the unknown conformation of CDRs (from new Fv sequences) by scoring the differences (disjoint combinations). New Fv sequences will henceforth be referred to as “Query” sequences, as they become the profiled object. The first step is, again, to number the Query Fv sequence and to assemble the respective IF sequence for each CDR to be predicted from the residues that correspond to the IF positions (defined previously during training). Subsequently, the DCP signature and the corresponding Target IF sequence set for each class of the corresponding CDR/length are loaded in turn. For each screened class, the signature signals are read one-by-one and the corresponding sets of IF fragment sequences are re-constructed. These sets of Target IF fragment sequences are then examined for disjointness versus the corresponding Query IF fragment sequence from the unknown CDR. If disjointness is observed between the Query fragment sequence and the Target fragment sequences in a given IF fragment (i.e., the Query fragment sequence is not in the list of Target fragment sequences), then the comparison score is increased by 1 and comparisons proceed with the next signature signal until all comparisons are performed. It is important to note again that signal matching is achieved by observing sequence differences (i.e., disjoint fragments) and not sequence similarities as is more common in the canonical model.

The final signature matching score (RDCPsignature) of a given class is equal to the comparison score (total number of disjoint signals), divided by the total number of signature signals in the DCP signature: (1) RDCPsignature=disjointsignalstotalsignaturesignals.

Once all classes in the given CDR/length are scored, the predicted conformation is the one with the RDCPsignature ratio closest to 1, and the workflow is repeated for the next CDR conformation to be predicted. A representation of the prediction workflow by DCP signatures can be seen in Fig. 3.

Figure 3 Representation of the workflow for CDR conformation prediction by DCP signatures.

New Fv sequences are referred to as “Query” sequences, as they become the profiled object, and therefore IF fragment sequences from the new Fv sequences become ‘Query IF fragment sequences’ for the purposes of prediction.

Canonical templates

The canonical templates were derived for every applicable conformational cluster, using the definitions of structurally-determining residues described in Martin & Thornton (1996). This choice was guided by the fact that the aforementioned study remains the most extended work on canonical residues, providing detailed tables of canonical templates for each conformational class.

Table 2 shows the canonical positions used for the creation of predictive templates in each applicable cluster, while the detailed canonical templates employed during blind-testing can be consulted in Supplemental Information 5. These templates were derived from the exact same training sequences used during DCP training, in order to allow a straight comparison between the two methods. It can be argued, that due to the nature of the level-1 clusters produced in Nikoloudis, Pitts & Saldanha (2014), the respective canonical templates may contain an unwarranted number of allowed residues, leading to misclassifications. This eventuality was explored by concurrently constructing, in selected cases (e.g., CDR-L3/9-residues, CDR-H1/13-residues), canonical templates from a small centralised portion of the cluster’s population, where conformation variations are minimal; namely those members that belonged to the cluster’s core. However, this training restriction led to an increased rate of misclassifications by canonical templates, probably because the sets of allowed canonical residues were not rich enough. For both this reason and for complete training conformity between the two methods, the exact same training sequences were used for DCP and canonical prediction from sequence.

Table 2 Canonical positions.

Table showing the canonical positions per CDR/length, used for CDR conformation prediction by canonical templates.

CDR/Length	Canonical positions	
CDR-L1/11	L2 L4 L25 L26 L28 L29 L30 L33 L34 L36 L46 L49 L51 L71 L90 L93	
CDR-L1/12	L2 L4 L25 L29 L33 L71 L90 L91 L93	
CDR-L1/13	L4 L25 L29 L30 L33 L66 L71	
CDR-L1/14	L4 L25 L29 L30 L31 L33 L66 L71 L90	
CDR-L1/15	L2 L4 L24 L25 L26 L28 L29 L30 L30c L33 L34 L51 L71 L90 L92 L93	
CDR-L1/16	L2 L4 L25 L26 L27 L29 L30a L30b L30c L30d L32 L33 L34 L51 L71 L90 L92 L93	
CDR-L3/8	L36 L89 L90 L91 L94 L95 L97 L98	
CDR-L3/9	L2 L3 L4 L28 L30 L31 L32 L33 L89 L90 L91 L92 L93 L94 L95 L96 L97 L98 H47	
CDR-L3/10	L4 L32 L36 L89 L90 L91 L92 L95a L96 L97 L98 H47	
CDR-H1/13	H2 H4 H20 H24 H26 H29 H32 H33 H34 H35 H48 H51 H69 H78 H80 H90 H94 H102	
CDR-H1/15	H20 H24 H26 H28 H29 H34 H48 H53 H78 H80 H94	
CDR-H2/9	H47 H51 H55 H59 H69 H71	
CDR-H2/10	H33 H47 H50 H51 H52 H53 H54 H55 H56 H58 H59 H69 H71 H78	
CDR-H2/12	L94 H47 H50 H51 H54 H55 H59 H69 H71 H78	

Sequence rules for CDR-H3-base prediction

Two sets of sequence rules for the prediction of the CDR-H3-base conformation were used: the first set from Shirai, Kidera & Nakamura (1999) and the updated set from Kuroda et al. (2008). The second set is an extension of the original set of rules based on examination of 314 new, non-redundant structures from the PDB. Blind-testing both sets of rules on the available test sets presented a good opportunity to examine their validity and, importantly, assess their extensibility by constant adaptation to new sequence findings. Although the respective publication was made in 2008, the updated set is referred to as “H3-rules 2007” in the corresponding text, so will henceforth be referred to accordingly.

Identification of multi-conformation full-rogue CDRs

During clustering, two conformational clusters that contain one or more members with identical CDR sequences were defined as ‘rogue’. For the DCP training and construction of canonical templates, it was also essential to search for, and deal with, structures that have the exact same Light and Heavy chain sequences within the clustering (training) dataset, but contain a CDR that belongs to different conformational clusters. These CDR structures were named ‘multi-conformation full-rogue CDRs’. Indeed, the presence of such CDRs in the training set would void DCP, as it would no longer be possible to detect any disjoint combinations between the sequences of the affected clusters. To a lesser degree, the same event would be detrimental for canonical predictions as well, since these full-rogue CDRs would have rogue templates, in the sense employed by Martin & Thornton (1996). However, as noted in North, Lehmann & Dunbrack (2011) and also observable in the detailed updated canonical templates (see Results section), the constantly increasing number of new antibody structures is already transforming most canonical templates into a ‘rogue’ status.

A visual examination of all detected occurrences was performed and detailed observations for Light and Heavy chain CDRs, and CDR-H3-base can be found in Supplemental Information 1. Based on these findings, it was decided to make no arbitrary exclusion of CDRs from the training set. The reason was that many rogue cases could warrant a dedicated study in order to make inferences on structure validity or potential conformational switches due to antigen/ligand contacts or backbone flexibility. Instead, it was decided that the affected clusters be merged into a combination of predictable conformations. In other words, affected clusters were treated as one during training for DCP and derivation of canonical templates. The implications of this choice are debated in the Discussion section. Finally, this identification of multi-conformational full-rogue members is presented as a piece of subsequent analysis based on the results of the complete clustering performed in Nikoloudis, Pitts & Saldanha (2014).

Validation of DCP training parameters

The DCP method allows selection of the CDR neighbourhood radius (IF) and the maximum combinatorial order of IF fragments. In this demonstration, IF radii of 4 Å, 6 Å and 8 Å (3 possible selections) were considered, as well as maximum orders up to triplets and up to quadruplets (2 possible selections). Therefore, DCP training per CDR/length was repeated for all 6 combinations of parameters and validated each time on the validation set. The combination of parameters that resulted in the higher predictive accuracy was retained for the final evaluation of the method on the test set. For the prediction of the CDR-H3-base conformation, quintets were also considered resulting in 3 additional training sessions. The selected parameters are listed in the Results section.

Blind-testing of sequence-based prediction methods for CDR conformation

Prediction results were categorised into four types: accurate, uncertain, false predictions and novel conformations. Predictions were considered failed in all cases other than the category “accurate”. As the prediction result from DCP signatures and canonical templates is based on the ratio of matched over the total number of signals/canonical residues, it is possible for two conformational classes to obtain the same maximum score. In these cases, the prediction is ‘uncertain’, and all classes with identical maximum score are output for reference. For an accurate prediction, the RMSD distance of the examined CDR conformation from a single cluster’s medoid was required to fall within that cluster’s radius. If this requirement was not matched, then the conformation was considered novel. In a few cases, the examined conformation appeared as an outlier between two clusters, displaying very similar RMSD distances to both their medoids; these outliers were also considered as novel conformations. Conformations with a CDR length with only one available cluster did not count towards any evaluation.

For the assessment of each method’s performance with regard to the prediction of the major cluster (class-I) in each CDR/length, the following measures are calculated: (2) Accuracy=TP+TNTP+TN+FP+FN

(3) Precision=TPTP+FP

(4) Recall=TPTP+FN

with TP, True Positive; TN, True Negative; FP, False Positive; FN, False Negative. Here, the positive class is the major conformation and the negative class refers to all the other conformations in that length. Therefore, ‘True Negative’ refers to the accurate prediction of a conformation other than the major in that length. Accordingly, ‘False Negative’ refers to the false prediction of a conformation other than the major one in the given length, while the actual conformation is the positive class.

Finally, as a technical appreciation of the combination of precision and recall, the F-measure is also provided: (5) F=2*Precision*RecallPrecision+Recall.

For the ‘uncertain’ predictions, with more than one class attaining equal maximum score, it was judged as more equitable to consider them as Negative results in all cases, since their predictive value is minimal in practice (i.e., the true conformation may be one or none of those reported). For those cases, if the true conformation of a CDR matches the major cluster in CDR/length, then that prediction counted as a False Negative for all further calculations — and as a True Negative in the case of the true conformation not matching the major class.

Post-evaluation DCP training and canonical templates’ updating

In order to evaluate the evolution in predictive accuracy of the different methods, an experiment was performed where both the training set and the validation set were combined and subsequently used for DCP training and canonical templates’ updating. The DCP parameters were retained from the previous validation step, meaning that parameters were not re-validated in this phase. Then, a final evaluation was performed on the test set. This stage was called ‘Phase 2’ and was analogous to a single cycle holdout experiment (Table 3). Phase-2 allowed an appreciation of the methods’ performances in time, as more antibody structures become available.

Table 3 Dataset naming and usage.

Summary of experiments performed, explaining the usage of datasets in each phase.

Dataset		Usage	
	Phase 1—initial evaluation	Phase 2—post-evaluation re-updating	
Clustering
set		- DCP training
- Canonical templates’ updating	- DCP training
- Canonical templates’ updating	
Blind set	Subset 1
(“validation set”)	- DCP validation
- Canonical templates’ testing
- Sequence rules’ testing	- DCP training
- Canonical templates’ updating	
Subset 2
(“test set”)	- DCP testing
- Canonical templates’ testing
- Sequence rules’ testing	- DCP testing
- Canonical templates’ testing	

Publicly available prediction tool

A GUI was developed with the Java Swing API for a computational tool that implements the prediction algorithms described for DCP and canonical templates (‘yCDRp’). The package (a jar file and a definitions folder) is available for downloading and use as a stand-alone desktop application at the “Humanisation bY Design” website, hosted by Birkbeck College, London at url: http://www.cryst.bbk.ac.uk/~ubcg07s. The GUI guides the user to manually structurally number their input Fv sequences and the tool’s initial release applies Phase-1 DCP signatures and canonical templates (‘definitions’) for CDR conformational prediction.

Results

Selected Interaction Frames for testing

Although during validation 3 IFs were assessed, in the following comparison of prediction results only the IF neighbourhood radius that gave the best predictive accuracy was considered. Table 4 shows the IFs that gave the best prediction results and their corresponding CDR neighbourhood radius. Positions at the end of the IF, marked as ‘n−x’, refer to CDR-H3 positions at a sequential distance x from the last residue n (H102). Since CDR-H3’s length is hypervariable, it was found that this notation better reflects the topological equivalence of numbered positions.

Table 4 Interaction Frames that resulted in the construction of the most accurate DCP signatures, and their respective CDR neighbourhood radius.

Notations ‘E’, ‘K’ and ‘K+’, at the end of the CDR-H3-base Interaction Frame, refer to the β-hairpin type that is favoured at the CDR-H3 apex, depending on the formation of an Extended (E), Kinked (K) and Kinked with double-bulged base (K+).

CDR	Interaction Frames	CDR Neighbourhood radius (Å)	
CDR-L1	L2 L3 L4 L5 L22 L24 L25 L26 L27 L28 L29 L30 L30a L30b L30c L30d
L30e L30f L31 L32 L33 L34 L36 L46 L48 L49 L50 L51 L52 L66 L67 L68
L69 L70 L71 L87 L89 L90 L91 L92 L93 L94 L95 H96 n-4 n-3	6	
CDR-L2	L30 L30a L30b L30c L30d L30e L30f L31 L32 L33 L34 L46 L47 L48 L49
L50 L51 L51a L51b L51c L51d L52 L53 L54 L55 L56 L57 L58 L59 L60
L61 L62 L63 L64 L65 L66 L67 L71 L72 L91 H32 H101 H102 n-5 n-4 n-3	4	
CDR-L3	L1 L2 L3 L4 L27 L28 L29 L30 L30a L30b L30c L30d L30e L30f L31 L32
L33 L34 L36 L49 L50 L87 L89 L90 L91 L92 L93 L94 L95 L95a L95b
L95c L95d L96 L97 L98 L99 H35 H45 H46 H47 H50 H58 H59 H60 H61
H95 n-6 n-5 n-4 n-3 n-2	4	
CDR-H1	L91 L92 L93 L96 H1 H2 H3 H4 H5 H6 H7 H20 H23 H24 H25 H26 H27
H28 H29 H30 H31 H31a H31b H31c H31d H31e H31f H31g H31h H31i
H31j H31k H32 H33 H34 H35 H47 H48 H49 H50 H51 H52 H52a H52b
H52c H52d H52e H52f H53 H56 H58 H69 H71 H72 H73 H74 H75 H76
H77 H78 H79 H80 H90 H91 H93 H94 H95 H96 H97 H98 H99 H100
H102 n-4 n-3 n-2	8	
CDR-H2	H24 H28 H29 H30 H31 H31a H31b H31c H31d H31e H31f H31g H31h
H31i H31j H31k H32 H33 H34 H35 H47 H48 H49 H50 H51 H52 H52a
H52b H52c H52d H52e H52f H53 H54 H55 H56 H57 H58 H59 H60 H61
H64 H68 H69 H70 H71 H72 H73 H74 H75 H76 H77 H78 H79 L94 L96	6	
CDR-H3-base	L34 L36 L43 L44 L45 L46 L49 L55 L87 L89 L91 L96 L98 H4 H27 H35
H37 H45 H47 H49 H91 H93 H94 H95 H96 H101 H102 n-3 n-2 E K K+	4	

Notations ‘E’, ‘K’ and ‘K+’, at the end of the CDR-H3-base IF, refer to the β-hairpin type that is favoured at the CDR-H3 apex, depending on the formation of an Extended E (Extended Negative EN and Extended Positive EP both resulting in the same β-hairpin ladder), single-bulged Kinked (K) or Kinked with double-bulged (K+) base. The hypothetical β-hairpin types (A/B/C/D) were derived from the definitions of the base type in Shirai, Kidera & Nakamura (1999). The profiling of an IF fragment that contains a hypothetical β-hairpin type would give the following correspondence in English: “is the co-existence of specific residues at specific Fv positions with a hypothetical β-hairpin type in CDR-H3 distinct within a class and therefore a disjoint event between different classes?” These categorical IF positions were introduced experimentally to the CDR-H3 IF and proved beneficial in practice. It was thus demonstrated that IFs may also include categorical features (another categorical example would be the CDR length) in order to allow the consideration of more complex combinations, for instance between residues and structural features.

Summary results for all experiments

Tables 5 and 6 show the accuracy of each method in each subset and experiment. Novel/non-previously clustered conformations observed in the new dataset are removed from the totals, in order to only assess performances on conformations that are predictable. Similarly, structures with a CDR length that contained less than 10 unique sequences in the clustered set were not considered. Canonical templates’ results show a reduced total test population in CDR-L3, because no templates were available for a length of 11-residues. Individual results are commented on later, per corresponding CDR.

Table 5 Individual accuracy percentages per experiment in CDR-L1 and -L3, excluding non-predictable (novel) conformations.

The previously acquired clustering set was used for initial DCP training and canonical templates’ updating. The newly downloaded blind dataset was divided in two subsets: for DCP, subset 1 was used for parameter validation (“validation set”), while subset 2 was used for evaluation (“test set”). Both subsets were used for evaluation of canonical templates, as no parameterisation was necessary, however the terms “validation” and “test” were retained for the two subsets for disambiguation and in order to allow direct comparisons. In post-evaluation Phase-2, the validation set was merged to the clustering set for DCP re-training and canonical templates’ re-updating. Updated methods were then evaluated on the test set that remained blind, but also were applied for retro-prediction on the validation set.

CDR-L1 predictions	
Phase-1 Initial DCP signatures	Phase-2 Updated DCP signatures	Phase-1 Initial canonical templates	Phase-2 Updated canonical templates	
Training:
clustering set
Evaluation:
validation set	Training:
clustering set
Evaluation:
test set	Training:
clustering + validation
sets
Evaluation:
validation set	Training:
clustering + validation
sets
Evaluation:
test set	Template Updating:
clustering set
Evaluation:
validation set	Template Updating:
clustering set
Evaluation:
test set	Training:
clustering + validation
sets
Evaluation:
validation set	Template Updating:
clustering + validation
sets
Evaluation:
test set	
99% (86/87)	99% (77/78)	100% (87/87)	98% (76/78)	92% (80/87)	96% (75/78)	98% (85/87)	96% (75/78)	
Cumulative evaluation on
validation + test sets	Cumulative evaluation on validation
+ test sets	Cumulative evaluation on validation
+ test sets	Cumulative evaluation on validation
+ test sets	
99% (163/165)	99% (163/165)	94% (155/165)	97% (160/165)	
CDR-L3 predictions	
Phase-1 Initial DCP signatures	Phase-2 Updated DCP signatures	Phase-1 Initial canonical templates	Phase-2 Updated canonical templates	
Training:
clustering set
Evaluation:
validation set	Training:
clustering set
Evaluation:
test set	Training:
clustering + validation
sets
Evaluation:
validation set	Training:
clustering + validation
sets
Evaluation:
test set	Template Updating:
clustering set
Evaluation:
validation set	Template Updating:
clustering set
Evaluation:
test set	Training:
clustering + validation
sets
Evaluation:
validation set	Template Updating:
clustering + validation
sets
Evaluation:
test set	
95% (84/88)	89% (70/79)	100% (88/88)	91% (72/79)	95% (69/73)	87% (62/71)	100% (73/73)	89% (63/71)	
Cumulative evaluation on
validation + test sets	Cumulative evaluation on validation
+ test sets	Cumulative evaluation on validation
+ test sets	Cumulative evaluation on validation
+ test sets	
92% (154/167)	96% (160/167)	91% (131/144)	94% (136/144)	

Table 6 Individual accuracy percentages per experiment in CDR-H1, -H2 and -H3, excluding non-predictable (novel) conformations.

Also see notes in Table 5.

CDR-H1 predictions	
Phase-1 Initial DCP signatures	Phase-2 Updated DCP signatures	Phase-1 Initial canonical templates	Phase-2 Updated canonical templates	
Training:
clustering set Evaluation:
validation set	Training:
clustering set
Evaluation:
test set	Training:
clustering + validation
sets
Evaluation:
validation set	Training:
clustering + validation
sets
Evaluation:
test set	Template Updating:
clustering set
Evaluation:
validation set	Template Updating:
clustering set
Evaluation:
test set	Training:
clustering + validation
sets
Evaluation:
validation set	Template Updating:
clustering + validation
sets
Evaluation:
test set	
96% (92/96)	95% (91/96)	100% (96/96)	96% (92/96)	79% (76/96)	83% (80/96)	83% (80/96)	85% (82/96)	
Cumulative evaluation on validation + test sets	Cumulative evaluation on validation
+ test sets	Cumulative evaluation on validation
+ test sets	Cumulative evaluation on validation
+ test sets	
95% (183/192)	98% (188/192)	81% (156/192)	84% (162/192)	
CDR-H2 predictions	
Phase-1 Initial DCP signatures	Phase-2 Updated DCP signatures	Phase-1 Initial canonical templates	Phase-2 Updated canonical templates	
Training:
clustering set
Evaluation:
validation set	Training:
clustering set
Evaluation:
test set	Training:
clustering + validation
sets
Evaluation:
validation set	Training:
clustering + validation
sets
Evaluation:
test set	Template Updating:
clustering set
Evaluation:
validation set	Template Updating:
clustering set
Evaluation:
test set	Training:
clustering + validation
sets
Evaluation:
validation set	Template Updating:
clustering + validation
sets
Evaluation:
test set	
93% (98/105)	87% (94/108)	100% (105/105)	81% (87/108)	58% (61/105)	57% (62/108)	64% (67/105)	56% (61/108)	
Cumulative evaluation on validation + test sets	Cumulative evaluation on validation
+ test sets	Cumulative evaluation on validation
+ test sets	Cumulative evaluation on validation
+ test sets	
91% (193/213)	90% (192/213)	58% (123/213)	60% (128/213)	
CDR-H3-base predictions	
Phase-1 Initial DCP signatures	Phase-2 Updated DCP signatures	1999 sequence rules	2007 sequence rules	
Training:
clustering set
Evaluation:
validation set	Training:
clustering set
Evaluation:
test set	Training:
clustering + validation
sets
Evaluation:
validation set	Training:
clustering + validation
sets
Evaluation:
test set	Evaluation:
validation set	Evaluation:
test set	Evaluation:
validation set	Evaluation:
test set	
89% (93/104)	91% (102/112)	100% (104/104)	88% (99/112)	83% (86/104)	87% (97/112)	86% (89/104)	84% (94/112)	
Cumulative evaluation on validation + test sets	Cumulative evaluation on validation
+ test sets	Cumulative evaluation on validation
+ test sets	Cumulative evaluation on validation
+ test sets	
90% (195/216)	94% (203/216)	85% (183/216)	85% (183/216)	

All predictions for every CDR in the test sets, along with a measure of RMSD distance of the Query conformation from the closest cluster medoid, can be consulted in detailed tables in Supplemental Information 3. Detailed tables with accuracy ratios per CDR/length, as well as extended statistics measuring the methods’ performance in predicting the major cluster in each CDR/length, are presented and commented on below (Tables 7 and 8 for CDR-L1/L3, Tables 9 and 10 for CDR-H1/H2 and Tables 11 and 12 for CDR-H3-base). In order to allow a direct comparison between methods, cumulative results (i.e., the entire new dataset) are considered in these detailed tables, but summaries of each method’s performance per subset and per experiment are also separately provided (Tables 5 and 6).

Table 7 Summary table of Phase-1 prediction results over all test data belonging to non-single cluster lengths, for CDR-L1 and -L3.

Percentages are rounded to the closest unit. Totals for canonical templates in CDR-L3 are marked in italics because they don’t include predictions for a length of 11-residues (no template available). For a direct comparison, total accurate predictions for DCP signatures for 8-, 9- and 10-resides CDR-L3 were 133/153 (87%). Totals include novel conformations.

	DCP signatures		Canonical templates			
CDR/Length	Accurately
predicted
conformations	Uncertain
predictions	False
predictions	Novel
conformations	Accurately
predicted
conformations	Uncertain
predictions	False
predictions	New CDR
sequences
in test set
(not present
in training set)	Unique CDR
sequences in
training set	
CDR-L1-11	97/107 (91%)	0/105	0/107	10/107 (9%)	93/107 (87%)	4/107 (4%)	0/107	68/107 (64%)	177	
CDR-L1-12	11/14 (79%)	0/14	2/14 (14%)	1/14 (7%)	8/14 (57%)	4/14 (29%)	1/14 (7%)	11/14 (79%)	25	
CDR-L1-13	16/17 (94%)	0/17	0/17	1/17 (6%)	16/17 (94%)	0/17	0/17	15/17 (88%)	26	
CDR-L1-14	10/10 (100%)	0/10	0/10	0/10	10/10 (100%)	0/10	0/10	6/10 (60%)	26	
CDR-L1-15	11/11 (100%)	0/11	0/11	0/11	10/11 (91%)	1/11 (9%)	0/11	9/11 (82%)	16	
CDR-L1-16	18/18 (100%)	0/18	0/18	0/18	18/18 (100%)	0/18	0/18	10/18 (56%)	71	
Total	163/177 (92%)	0/177	2/177 (1%)	12/177 (7%)	155/177 (88%)	9/177 (5%)	1/177 (0.5%)	119/177 (67%)	341	
CDR-L3-8	18/19 (95%)	0/19	1/19 (5%)	0/19	17/19 (89%)	1/19 (5%)	1/19 (5%)	12/19 (63%)	44	
CDR-L3-9	111/119 (93%)	1/119 (1%)	6/119 (5%)	1/119 (1%)	110/119 (92%)	4/119 (3%)	4/119 (3%)	84/119 (71%)	359	
CDR-L3-10	4/15 (27%)	0/15	4/15 (27%)	7/15 (47%)	4/15 (27%)	2/15 (13%)	2/15 (13%)	14/15 (93%)	26	
CDR-L3-11	19/25 (76%)	0/25	1/25 (4%)	5/25 (20%)	N/A	N/A	N/A	23/25 (92%)	36	
Total	152/178 (85%)	1/178 (1%)	12/178 (7%)	13/178 (7%)	131/153 (86%)	7/153 (5%)	7/153 (5%)	133/178 (75%)	465	

Table 8 Extended performance measures for major cluster (class-I) predictions in each CDR-L1 and -L3 length (Phase-1).

No canonical templates were available for CDR-L3/11-residues. The asterisk points the fact that clusters in CDR-L3/10-residues are all small, however cluster CDR-L3-10-I was technically considered here for consistency with all other major clusters.

	Class-I predictions	Statistics	
CDR/Length	True positives	True negatives	False positives	False negatives	Accuracy	Precision	Recall	F-measure	
	DCP signatures	
CDR-L1-11	82	21	4	0	0.96	0.95	1.00	0.98	
CDR-L1-12	7	5	1	1	0.86	0.88	0.88	0.88	
CDR-L1-13	18	0	1	0	0.95	0.95	1.00	0.97	
CDR-L1-14	8	2	0	0	1.00	1.00	1.00	1.00	
CDR-L1-15	11	0	0	0	1.00	1.00	1.00	1.00	
CDR-L1-16	18	0	0	0	1.00	1.00	1.00	1.00	
	Canonical templates	
CDR-L1-11	82	21	4	0	0.96	0.95	1.00	0.98	
CDR-L1-12	4	5	0	5	0.64	1.00	0.44	0.62	
CDR-L1-13	18	1	0	0	1.00	1.00	1.00	1.00	
CDR-L1-14	8	2	0	0	1.00	1.00	1.00	1.00	
CDR-L1-15	10	0	0	1	0.91	1.00	0.91	0.95	
CDR-L1-16	18	0	0	0	1.00	1.00	1.00	1.00	
	DCP signatures	
CDR-L3-8	14	4	1	0	0.95	0.93	1.00	0.97	
CDR-L3-9	107	4	8	0	0.93	0.93	1.00	0.96	
CDR-L3-10*	1	11	0	3	0.80	1.00	0.25	0.40	
CDR-L3-11	19	1	5	0	0.80	0.79	1.00	0.88	
	Canonical templates	
CDR-L3-8	13	4	1	1	0.89	0.93	0.93	0.93	
CDR-L3-9	105	8	4	2	0.95	0.96	0.98	0.97	
CDR-L3-10*	1	11	0	3	0.80	1.00	0.25	0.40	
CDR-L3-11	–	–	–	–	–	–	–	–	

Table 9 Summary table of Phase-1 prediction results over all test data belonging to non-unique-cluster lengths, for CDR-H1 and -H2.

Totals include novel conformations.

	DCP signatures		Canonical templates			
CDR/Length	Accurately
predicted
conformations	Uncertain
predictions	False
predictions	Novel
conformations	Accurately
predicted
conformations	Uncertain
predictions	False
predictions	New CDR
sequences in
test set
(not present in
training set)	Unique CDR
sequences
in training set	
CDR-H1-13	178/201 (89%)	0/201	8/201 (4%)	16/201 (8%)	153/201 (76%)	24/201 (12%)	8/201 (4%)	135/201 (67%)	419	
CDR-H1-15	5/9 (56%)	0/9	2/9 (22%)	2/9 (22%)	3/9 (33%)	2/9 (22%)	2/9 (22%)	8/9 (89%)	27	
Total	183/210 (87%)	0/210	10/210 (5%)	18/210 (9%)	156/210 (74%)	26/210 (12%)	10/210 (5%)	143/210 (68%)	446	
CDR-H2-9	41/41 (100%)	0/41	0/41	0/41	27/41 (66%)	14/41 (34%)	0/41	30/41 (73%)	117	
CDR-H2-10	145/168 (86%)	6/168 (4%)	14/168 (8%)	3/168 (2%)	89/168 (53%)	60/168 (36%)	16/168 (10%)	128/168 (76%)	350	
CDR-H2-12	7/8 (88%)	0/8	0/8	1/8 (13%)	7/8 (88%)	0/8	0/8	4/8 (50%)	39	
Total	193/217 (89%)	6/217 (3%)	14/217 (6%)	4/217 (2%)	123/217 (57%)	74/217 (34%)	16/217 (7%)	162/217 (75%)	506	

Table 10 Extended performance measures for major cluster (class-I) predictions in each CDR-H1 and -H2 length (Phase 1).

	Class-I predictions	Statistics	
CDR/Length	True positives	True negatives	False positives	False negatives	Accuracy	Precision	Recall	F-measure	
	DCP signatures	
CDR-H1-13	173	6	21	1	0.89	0.89	0.99	0.94	
CDR-H1-15	4	1	4	0	0.56	0.50	1.00	0.67	
	Canonical templates	
CDR-H1-13	153	7	20	21	0.80	0.88	0.88	0.88	
CDR-H1-15	3	2	3	1	0.56	0.50	0.75	0.60	
	DCP signatures	
CDR-H2-9	41	0	0	0	1.00	1.00	1.00	1.00	
CDR-H2-10	103	45	4	16	0.88	0.96	0.87	0.91	
CDR-H2-12	7	0	1	0	0.88	0.88	1.00	0.93	
	Canonical templates	
CDR-H2-9	27	0	0	14	0.66	1.00	0.66	0.79	
CDR-H2-10	74	43	6	45	0.70	0.93	0.62	0.74	
CDR-H2-12	7	0	1	0	0.88	0.88	1.00	0.93	

Predictions for CDR-L1

For the DCP method in CDR-L1 predictions, validated training parameters were a 6 Å IF and a combinatorial order up to triplets. The entire clustered dataset was used during training (excluding outlying members in order to reduce the complexity of the predictable testing structure). The method achieved an overall ratio of accurately predicted CDR-L1 of 92% (163/177), while the number of novel conformations in the test set, represented another 7% (12/177; Table 7). Of special note is the fact that in 12- and 13-residue CDR-L1 lengths, classified as non-predictable or statistically uncertain (‘type III’) in North, Lehmann & Dunbrack (2011), the method predicted successfully 79% and 94% of the test CDRs respectively (11/14, 1/14 being a novel conformation, and 16/17, 1/17 being novel, respectively). This becomes more striking when considering that 79% (11/14) and 88% (15/17) respectively of the test CDR sequences were new and not represented in the training set.

Predictions with canonical templates were overall accurate in 88% (155/177) of the test set (Table 7). This prediction score was mainly lowered by the existence of a number of uncertain predictions (∼5%, 9/177), where more than one conformation achieved the same maximum canonical matching. As far as the prediction performance towards the major cluster in each length is concerned (level-1 clusters: class-I; Nikoloudis, Pitts & Saldanha, 2014), DCP signatures proved superior to or as effective as canonical templates in most measures, except for the precision in 12-residue CDR-L1 (0.88 vs. 1.0); and also accuracy (0.95 vs. 1.00) and precision (0.95 vs. 1.00) in 13-residue CDR-L1 (Table 8). In both lengths, this result is due to the fact that canonical templates output an uncertain prediction in actual conformations other than the one corresponding to the major cluster in that length (i.e., not a class ‘-I’ tag), which technically counted as True Negatives in our assessment. These True Negatives increased the respective accuracy and precision scores for the canonical model. In contrast, canonical templates scored very low in recall in 12-residue CDR-L1 (0.44) vs. DCP (0.88). Concerning the most voluminous cluster in CDR-L1, i.e., L1-11-I (160/434 or ∼37% of total CDR-L1 clustered sequences; Nikoloudis, Pitts & Saldanha, 2014), both methods performed equally well with an F-measure of 0.98 (Table 8), while 65% (70/107) of the test CDRs were new sequences (Table 7). Thirty-three structures had a CDR-L1 length where only one cluster was available (7, 10 and 17 residues), or less than 10 unique sequences were available during clustering (9 residues). Two structures had a CDR-L1 with a new, non-clustered length (8 residues).

After discarding non-predictable CDRs (novel conformations, very low clustered populations, or only one cluster per length), DCP signatures achieved an overall accuracy of 99% (163/165), as compared to 94% (155/165) for canonical templates (Table 2). Although both methods performed very well, DCP signatures’ performance proved slightly superior in all individual assessments. However, post-evaluation DCP training and re-assessment (Phase-2) on the test set, resulted in two wrong and one correct prediction switches, resulting in a roughly 1% lowering of the final accuracy of the method (Table 5; comparison between “Initial” and “Updated DCP signatures, Evaluation: test set”). Accordingly, post-evaluation updating of canonical templates didn’t have any effect on the predictions obtained for the new dataset (96% in both cases). Results suggest stability of both methods in view of the new structures, although assessment with bigger test sets will be required in the future for a safer conclusion.

Predictions for CDR-L2

Nearly all 178 new structures with a Light chain had a CDR-L2 belonging to cluster L2-7-I (175/178, ∼98%). This was expected, since over 96% (272/282) of the clustered CDRs had this conformation (Nikoloudis, Pitts & Saldanha, 2014). Moreover, conformational differences between the three observed clusters are rather minimal (mostly peptide flips that additively result in inter-cluster medoid distances ranging between 1.1 Å and 1.36 Å) and can be marginally characterised as variants of the main L2-7-I conformational theme. For these reasons, the predictive methods were not applied in CDR-L2. Future work targeting, with greater precision, the variants within a cluster could reveal whether these conformational differences are predictable by sequence alone, or even the result of experimental errors.

Predictions for CDR-L3

For the construction of CDR-L3 DCP signatures, a 4 Å IF was used and for detection of disjoint signals, IF fragment combinations were examined up to triplets. The DCP method achieved an overall ratio of accurately predicted conformations of 85% (152/178), while 7% (13/178) of the test set were novel conformations (Table 7), of which 12/13 had new CDR sequences (Supplemental Information 3). The lowest accuracy was observed in 10-residue CDR-L3; a length that, interestingly, seems hypervariable in conformation. From the 27 unique sequences of CDR-L3/10-residues in the clustering dataset, 12 clusters were formed each containing between 1 and 4 unique CDR sequences, while 6 more CDRs were labelled as outliers/singletons (Nikoloudis, Pitts & Saldanha, 2014). Nonetheless, since 7/15 test structures with CDR-L3/10-residues were novel conformations (all with new CDR sequences; Table 7) it was impractical to make any conclusions regarding predictive efficacy in this CDR length. The biggest cluster in CDR-L3 is L3-9-I containing 328/480, or ∼68%, of the clustered, non-redundant structures, all CDR-L3 lengths considered (Nikoloudis, Pitts & Saldanha, 2014); a percentage that is interestingly verified in the present new dataset (119/178, or 67% of the unique CDR-L3 sequences, Table 7). For this cluster, the DCP method achieved an accuracy of 0.93, while its F-measure was 0.96 (Table 8).

The canonical model achieved comparable overall prediction accuracy of 86% (131/153), excluding predictions for CDR-L3/11-residues as no template was available (Table 7). For a direct comparison, total accurate predictions for DCP signatures for 8-, 9- and 10-residue CDR-L3 were 133/153 (87%). Canonical model’s score was lowered, for this method as well, mainly by the presence of many novel conformations in CDR-L3/10-residues, and also a few uncertain predictions obtained in CDR-L3/8- and 9-residues. Overall, both methods performed equally well, with an only marginal superiority of the DCP method in CDR-L3/8- and 9-residues. Canonical templates also achieved a slightly better score in accuracy (0.95 vs. 0.93) and precision (0.96 vs. 0.93) of the major class-I in CDR-L3/9-residues (Table 8), which is again due to the fact that uncertain predictions output for non-class-I conformations technically counted as True Negatives. No predictions were made for a length of 5-residues as only one cluster was available, and for 12- and 13-residues as less than 10 unique CDR sequences were available in the clustered data (6 and 3, respectively; Nikoloudis, Pitts & Saldanha, 2014).

Initial cumulative performances after removing non-predictable conformations were comparable: 92% (154/167) for DCP signatures and 91% (131/144) for canonical templates (Table 5), although DCP was applied in one additional length (11-residues). In all individual assessments DCP performed equally or slightly better, while both methods took equal benefit from post-evaluation training/updating (Phase-2), gaining roughly 2% in overall accuracy (Table 5, comparison between “Initial” and “Updated DCP signatures/canonical templates, Evaluation: test set”).

Predictions for CDR-H1

For the construction of DCP signatures for the CDR-H1 prediction, training parameters were the following: an 8 Å IF and fragment combinations up to triplets. Clustered populations in CDR-H1 lengths 13- and 15-residues represented 96% (446/465) of the total non-outlying, non-redundant CDR-H1 population (Nikoloudis, Pitts & Saldanha, 2014) and are the only ones that formed more than one conformational cluster. The DCP method achieved an overall ratio of correct predictions of 87% (182/210), while 9% (18/201) were novel/non-clustered conformations (Table 9), most of which were observed in 13-residue CDR-H1. The method’s performance was rather poor in 15-residue CDR-H1 (56%, 5/9 accurate predictions); although the small number of test structures in this length doesn’t allow any concrete conclusion. It is notable that two out of three structures with H1-15-II conformations (3TJE, 3THM, Supplemental Information 3) were not predicted correctly — all 3 represented by new CDR-H1 sequences. A possible reason for this could be the small training population for the 3 clusters in CDR-H1/15-residues (24/2/1 unique sequences, respectively; (Nikoloudis, Pitts & Saldanha, 2014). In CDR-H1/13-residues, 88% (177/201) were accurately predicted, 4% (8/201) were false predictions, and 8% (16/201) were novel conformations (Table 9).

Generated canonical templates for CDR-H1 displayed an increased degree of allowed residues’ overlap. This was expressed by an increased number of uncertain predictions (maximum score by more than one template): 26/210, or 12% (Table 9). The overall ratio of accurate predictions was 74% (156/210), while false prediction represented approximately 5% (10/210) (Table 9). Comparing the performance of both methods in predicting the major clusters in each CDR-H1 length, the DCP method was from marginally to significantly superior to the canonical model in all measures (Table 10).

After removal of non-predictable conformations, DCP signatures achieved a cumulative accuracy of 95% (182/192) as opposed to the canonical model with 81% (156/192; Table 6). The performance of DCP signatures was accordingly superior in all individual assessments, while post-evaluation training/updating benefited both methods by ∼1%–2% (Table 6, comparison between “Initial”and “Updated DCP signatures/canonical templates, Evaluation: test set”).

Predictions for CDR-H2

Predictions for CDR-H2 conformation concerned three lengths, where there were more than one cluster and more than 10 unique clustered CDR sequences: 9-, 10-, and 12-residue CDR-H2. Length 10-residues was of additional interest as it represented the only case, all CDRs considered, that featured both a considerable total population (350 unique clustered sequences) and two well-populated clusters with an approximate 1:2.5 ratio in non-redundant members. For the construction of DCP signatures, a 6 Å IF was employed and fragments up to quadruplets were compared. The method achieved accurate predictions in 89% of the test CDRs (193/217), made an uncertain prediction in 6 cases (∼3%), and a false prediction in 14 cases (∼6%), while 4 more CDRs were novel conformations (∼2%, Table 9). For CDR-H2 lengths 9- and 10-residues alone, North, Lehmann & Dunbrack (2011) reported a theoretical percentage of correct predictions of 80%, consulting extensively the identity of the residue at position H71, and of 78%, using hidden Markov models.

Canonical templates for CDR-H2 displayed a very pronounced degree of overlap between allowed residues, which was even more severe than was observed in CDR-H1. This caused the percentage of uncertain predictions to rise to 34% (74/217), while the false predictions were 7% (16/217). The overall accuracy was therefore only 57% (123/217, Table 9). Canonical performance in predicting the major cluster in each length suffered accordingly, although not as dramatically as the global accuracy would suggest (F-measure: 0.79 for CDR-H2/9-residues, 0.74 for CDR-H2/10-residues, Table 10).

In the case of CDR-H2 then, the advantage of comparing combinations of residues (DCP) was observed in a more prominent manner. Therefore, observations here support the initial hypothesis of the degree of sequence-to-structure residue synergistic complexity and the non-linear determination of conformation by local and neighbouring residue preferences. However, post-evaluation DCP training resulted in the severe reduction of signals in the signatures of clusters H2-10-I and H2-10-II. This led to an increased rate of uncertain predictions during re-evaluation with the updated signatures, which was reflected by a 6% loss in accurate predictions (Table 6, comparison between “Initial” and “Updated DCP signatures, Evaluation: test set”), although only less than 1% loss when comparing the respective cumulative results (Table 6). Re-updating of canonical templates, on the other hand, resulted expectedly in a slight decrease in accuracy (∼1%), as updating could only accentuate the existing template overlap effect; although cumulatively, re-updating increased the ratio of accurate predictions by ∼2% (Table 6). Normally, this behaviour of DCP would suggest the need for re-parameterisation, using e.g., an increased order of combinations (e.g., quintets) in order to preserve predictive performance in time. Nonetheless, since the difference in global performance of the two methods is already so dramatically in favour of the DCP method (Tables 6 and 9), it was judged preferable to demonstrate rather than attenuate this effect, as a useful piece of critical assessment for this new method that will allow improved future implementations.

Predictions for CDR-H3-base conformation

The pronounced sequence, length and conformational hypervariability in CDR-H3 was verified during the clustering (Nikoloudis, Pitts & Saldanha, 2014) and in this landscape of variability it was completely impractical to apply predictive DCP on the complete CDR-H3 conformation, at least not in the form of the current implementation of this new method; a remark also arising from the earliest conception of the canonical model. Nonetheless, as a major advance in the prediction of CDR-H3 conformation from sequence concerns the formulation of sequence rules for the CDR-H3-base, DCP was applied on three CDR-H3-base categories: Kinked (K), Extended negative (EN) and Extended positive (EP). Prediction of the double-bulged Kinked base (K+), was not attempted on this occasion for simplicity.

For DCP signatures, a 4 Å IF was used and compared fragments up to quintets were compared. The DCP method made 195/216 correct predictions (90%) for the CDR-H3-base conformation. Comparatively, application of sequence rules resulted in 183/216 (85%) accurate predictions, for both the 1999 (Shirai, Kidera & Nakamura, 1999) and 2007 (Kuroda et al., 2008) sets of rules respectively (Table 11). More specifically, the updated set of rules resulted in 11 correctly switched predictions (∼5%) and 13 falsely switched predictions (∼6%, two switches were from a wrong prediction to another wrong prediction, the rest were from correct to wrong); 21 incorrect predictions made by the original set were retained in the 2007 set (∼10%; Supplemental Information 3).

Table 11 Summary table of Phase-1 prediction results for the CDR-H3-base conformation over all test data.

	DCP signatures	H3-rules, 1999 edition	H3-rules, 2007 edition	
	Accurately
predicted
conformations	False
predictions	Accurately
predicted
conformations	False
predictions	Accurately
predicted
conformations	False
predictions	
CDR-H3-base
conformation	195/216 (90%)	21/216 (10%)	183/216 (85%)	33/216 (15%)	183/216 (85%)	33/216 (15%)	

The methods’ performances were evaluated separately in predicting the Kinked base, which represents the most frequent base conformation (roughly 7:2:1 ratio between K–EN–EP conformations in all datasets combined). The updated rule set presented an almost identical performance, over all measures, to the original set (Table 12). It therefore cannot be verified that the updating of sequence rules on the basis of new structures is beneficial; it can be argued that a point may appear where the rules’ predictive performance may no longer warrant their increasing complexity. In comparison, predictions with DCP signatures brought a slight improvement over both sets of sequence rules, all measures considered (accuracy 0.90 vs. 0.85, F-measure 0.95 vs. 0.92, Table 12). Although this improvement is still marginal, it confirms the new method’s consistency in out-performing, or performing at least as well as, the existing methods in all CDRs including CDR-H3.

Table 12 Extended performance measures for Kinked base predictions in CDR-H3 (Phase 1).

	Class-I predictions	Statistics	
	True positives	True negatives	False positives	False negatives	Accuracy	Precision	Recall	F-measure	
CDR-H3, kinked
base conformation	DCP signatures	
191	4	13	8	0.90	0.94	0.96	0.95	
H3-rules, 1999 edition	
182	1	15	18	0.85	0.92	0.91	0.92	
H3-rules, 2007 edition	
178	5	13	20	0.85	0.93	0.90	0.92	

Discussion

The historical approach used for CDR prediction from sequence alone relies on canonical templates or in simpler cases the existence of a single conformational cluster for a given CDR length. For the hypervariable CDR-H3, where sequences, lengths and conformations show great diversity, sequence rules were formulated in order to allow the prediction of only the base of the loop. While the canonical model has been, and still is, effective in predicting a number of CDR conformations, its strength is inevitably weakened as more antibody structure become available. As the construction of canonical templates consists of identifying structurally-determining residues at specific positions that are exclusive to each canonical class, the proliferation of sequences in CDR clusters gradually creates overlapping, or rogue, templates (not to be confused with the multi-conformation, full-rogue CDRs in this work). This was first observed by Martin & Thornton (1996) and was acknowledged by North, Lehmann & Dunbrack (2011) where canonical templates discreetly gave way to statistical consensus sequences.

A typical problem with canonical templates, and by extension with statistical consensus sequences, is that they require the presence of previously observed residues in specific positions, without the consideration that certain overlapping combinations of residues may render the targeted CDRs unpredictable. While this could be statistically acceptable in the past as far as positive predictability was concerned, the great increase of CDRs in the PDB results in conformational clusters with highly overlapping canonical templates or consensus sequences; for example in CDR-H2/10-residues, all 14 canonical positions contain at least one overlapping residue between one or more other templates of the same length (Supplemental Information 5). Additionally, another fact that becomes prominent with richer datasets is that many CDR conformations do not depend solely on their own sequence but receive structurally-determining influence from the antibody’s framework (Tramontano, Chothia & Lesk, 1990; Martin & Thornton, 1996; Morea et al., 1997; North, Lehmann & Dunbrack, 2011). These problems can sometimes be dealt with by application of Hidden Markov Model (HMM) analysis. However this requires a considerable number of cluster members for the model to remain reliable, and to some extent removes the simplicity that made the canonical model attractive in predicting the conformations of antibody CDRs. Nonetheless, sequence logos were constructed for the training clusters using Berkeley’s WebLogo facility (http://weblogo.berkeley.edu/logo.cgi) (Crooks et al., 2004) and are provided as Supplemental Information 4.

As the present predictive methods essentially predict a class of similar conformations and not the actual CDR coordinates, their utility as far as modelling the loop is concerned is to be complemented by the cluster and sequence characteristics tied to the predicted class. Since the present work was based on the clustering analysis presented in Nikoloudis, Pitts & Saldanha (2014), after a positive prediction, the members/structures corresponding to the cluster’s (level-1) medoid and extremities should be extracted in order to get a quick appreciation of the extents of possible (known) conformational variability in the Query structure. A multiple alignment within that cluster’s sequences would result in the members most similar to the Query sequence, one of which would form a preliminary template. Depending on the selection of the most appropriate template and any other CDRs to be modelled, a possible conformational shift towards the core or the extremities of the cluster’s level-1 conformational theme would be inferred. Furthermore, consultation of other sequences of members belonging to the same level-2 or even level-3 cluster as the template, would reveal if specific sequence differences observed in the Query sequence are allowed within that specific variant. Finally, sequence features observed at the daughter-level could potentially also guide the modelling of the subtle conformational characteristics of that sub-cluster (i.e., at levels 2/3) with regard to the main/parent conformational theme that was predicted (i.e., at level-1).

Based on the present prediction results, a conclusion that can be drawn regarding the canonical model is that it still presents an acceptable predictive capability, at least in most Light chain CDR lengths. Overall, accurate predictions by canonical templates were 565/757 (74.6%) in CDR-L1, -L3, -H1 and -H2, with 47/757 (∼6%) being novel, non-predictable conformations (sum of results in Tables 7 and 9); after removal of non-predictable conformations, total cumulative accurate predictions were then 565/710 or 79.6% (sum of results in Tables 5 and 6). Its performance in Heavy chain CDRs though, where the overlap of canonical templates resulted in important accuracy loss (Tables 6 and 9), could suggest that the efficacy of the canonical model may be bound to decrease over time as more structures become available. One possible solution for retaining the practicality of the canonical model could be a k-fold cross-validation analysis of a dataset in order to obtain the canonical templates that best predict the available conformations; and then keep those templates locked until the assessed performance of the model begins to decline again in the future. Alternatively, the re-sampling of established canonical positions in each CDR length could also potentially result in better performance, i.e., a cross-validation analysis with reduced sets of canonical positions. Such a process is expected to virtually remove several heavily overlapping positions and allow better template specificity.

Following a much more supervised approach, sequence rules used for prediction of CDR-H3 features still demonstrate a satisfactory predictive potential as confirmed by the blind-testing sessions. With an overall accuracy of 85% during testing over the two sets of rules, it can be supported that the sequence basis for the CDR-H3-base conformation is essentially acquired. On the other hand the addition of 8 new rules or rule-adaptations in the updated set, on top of the original 4, didn’t procure an improvement in accuracy. It can be argued that the test set of 216 sequences was relatively small for safe conclusions, compared to the 311 sequences used during the formulation of the updated rules. However, these additional rules were created for the correct identification of only 47/311 (15%) of CDR-H3-bases that were misclassified by the original set of rules (Kuroda et al., 2008). Interestingly, exactly the same percentage of misclassified bases (15%; Table 11) was again also displayed by the updated rules during the testing session, suggesting a possible attained limit in the efficacy of the sequence rules. Moreover, false switches from the original set’s prediction were not avoided (13 cases), while the correct prediction switches were fewer than the number of false predictions retained from the original set (11 and 21, respectively in Supplemental Information 3). Therefore, the test set was generally representative of the predictive challenge a researcher may encounter and, as previously mentioned, that sequence rules could already have reached a point where their further specialisation towards improvement of accuracy has become impractical, ineffective, or both.

The newly proposed predictive method (DCP) achieved an overall score of correct predictions in all examined CDRs of 88.7% (885/998), while approximately 5% (47/998) of the test CDRs represented novel, unpredictable conformations (sum of results in Tables 7, 9 and 11); after removal of non-predictable conformations, total cumulative accurate predictions were then 885/951 or 93.1% (sum of results in Tables 5 and 6). The improvement over the canonical model or sequence rules was consistent in all CDRs, ranging from 1% in CDR-L3 to 33% in CDR-H2 (average 10.8%, median 6%) cumulatively over the entire new dataset (comparison between cumulative results in Tables 5 and 6), and ranging from 2% to 30% (average 9.7%, median 5.5%) over the test subset only (initial evaluation of test set, Tables 5 and 6). This improvement was verified during the evaluation of prediction performance for the most populated, and thus statistically most important, cluster in each predictable length. Over 60 total measures (15 common categories, 4 statistical evaluations per category), the DCP method’s score ranged from equal to significantly better in all but 6 cases, in which canonical templates performed marginally better mainly due to technicalities of the assessment that were discussed previously (Tables 8, 10 and 12). With all but two F-measure scores (L3/10-residues, H1/15-residues) being better than 0.88 (average 0.90, median 0.96), confidence for accepting or rejecting the adoption of the major conformation in length by the unknown CDR can be relatively high.

This performance was deemed encouraging, considering the method’s novel and embryonic nature. It can therefore also be argued that the threshold-free approach of the initial clustering was advantageous for prediction as it created richer clusters by including more sequence examples and possible variants of a conformational theme. These variants could have diminished the predictive efficiency of the assessed methods, if considered as separate clusters in the first place. Indeed, unless these variants were later detected as multi-conformation full-rogues which would lead to their predictive merging, their separation from the main conformational theme would produce poorer training/updating results due to considerably fewer examples per profiled cluster. In any case, it would also be interesting to apply DCP at levels -2 and -3 of the nested initial classification, in order to explore the potential of prediction of the more subtle variants, which would be of increased importance to antibody engineering, if successful. Moreover, future dedicated work on DCP signatures may bring further improvements in the overall predictive potential by proposing more elegant implementations than the basic approach employed in this work.

Clusters that contained members with the exact same Fv sequences were merged for training/updating and prediction. Hence, DCP signature or canonical template matching of a combined predictable conformation reported all the affected conformations at the same time. In these cases there was inevitably a loss of specificity towards the prediction of each separate conformation. However in practice, in 295/301 (∼98%) related cases of accurate prediction by DCP signatures of a combination of clusters, the true conformation was always that of the major class of the set. This could suggest that those smaller clusters that contain multi-conformation full-rogue CDRs are more valuable for merely being part of the known conformational repertoire of that CDR and for becoming the object of sequence-to-structure and/or CDR induced-fit studies, rather than representing important predictable conformations.

Alternatives to the above approach for successful training would be to exclude the sequences of all involved members from their respective cluster sets, or to exclude the sequence sets of the smaller cluster altogether as not important for prediction; both scenarios hiding potential training inconveniences. Therefore, the predictive cluster merging preserves the availability of sequence information, did not practically reduce prediction sensitivity and presented no obvious bias toward one of the two predictive methods that are compared for each CDR. On the contrary, most merges may be pointing to closely related conformations whose divergence is due to external factors, in which case it makes more sense to consider them in a combined fashion.

Perhaps the biggest future challenge for the DCP method would be to detect the presence of a novel class – not the novel conformation itself – but merely the potential to avoid a false positive identification. This is an inconvenience shared by all sequence-based methods, since they always attribute a class to an unknown structure. The avoidance of false positives (all classes considered) could be achieved in time as signatures become more specific, in which case a positive identification would require a ratio score better than a defined threshold (e.g., no positive prediction below RDCPsignature < 0.5, Eq. (1)). Alternatively, this could be achieved with the definition of a negative class. The training protocol of the DCP method may indeed allow for such a process, precisely because it is searching for differences between the compared IF sequence sets instead of similarities. An exploratory approach could be the selective mixing of different classes divided between Query and Target sets, in order to represent a non-existent conformation or combinatorial chimera, for profiling of disjoint combinations. Signatures obtained from such training should then be tested for positively attracting unknown conformations, without interfering with known classes.

The biggest culprit during DCP training was undoubtedly its execution complexity which scales in factorial time. In practice with a short 4 Å IF, single-threaded execution time was acceptable for DCP with IF sequence fragments up to quintets, or even sextets (i.e., up to 2–3 min per CDR). However with longer (up to 8 Å) IFs, execution time becomes very quickly prohibitive, with quartets’ training requiring sometimes close to 50 min per CDR on the available computational setup (2.67 MHz Intel i5 quad-Core processor). As was revealed by the test results, supervised exploration of a number of selected [IF length]/[fragment order] combinations of training sets proved sufficient in order to reach and surpass the performance of the other established methods. However for optimisation of DCP signatures, a k-fold cross-validation of the signature signals may be required, which will be the focus of a future study. Of course, it cannot be ruled out that future dedicated studies may also propose a more efficient training procedure, e.g., by defining shorter IFs based on a criterion other than the structural neighbourhood of a CDR. Also, another way for producing more accurate and specific DCP signatures could be in the statistical validation of the disjoint combinations/events. Toward this end, a probabilistic closed-form equation for selecting only statistically significant signature signals is proposed as an appendix in Supplemental Information 2. As a final suggestion, the representation of IF sequence sets using reduced non-overlapping amino acid alphabets is another intriguing possibility to be explored for an improved implementation of the DCP method.

It should be noted that both DCP signatures and canonical templates are by design able to achieve a maximum score with the totality of a training set, but with DCP an uncertain retro-prediction is not possible. After post-evaluation training/updating on both the clustering and the validation sets, re-evaluation of the validation set showed superior aptitude of DCP signatures in retro-predicting the set they were trained upon (100% correct predictions, Tables 5 and 6). This behaviour was expected as DCP signatures capture all the combinatorial differences between one class and all the others. Additionally, the IF sequence of any Query structure gets included into the Target IF sequence sets of all clusters except for the one that corresponds to the Query structure. Hence, no disjointness can be observed between the Query IF sequence fragments and any non-corresponding cluster, so uncertain predictions are essentially avoided. In contrast, canonical templates display a more linear ensemble of intra-class similarities that become overlapping, which penalises the predictive accuracy of the model. This means that, at least in theory, updating of DCP signatures by adding new sequences to the training set should produce more stable and accurate predictive models. Thus, provided that an optimised set of training parameters (IF radius and combinatorial order) is acquired, disjoint signals should become naturally filtered and signatures should be increasingly specific to each class, as more structures and their sequences bring additional examples of clustered conformational themes. Analysis of the individual signals within these increasingly specific signatures could then potentially assist in discovering important interactions that contribute to CDR conformation; a feature that is not easily accessible in other analytical methods such as HMM or neural networks.

As a random example of a possibly interesting combination of residues from the obtained signatures of cluster L1-11-II, a prominent detected disjoint couplet that is retained in phase-2 signatures and also potentially validated as significant by absolute probabilistic significance (p = 0.0012) was L5/L66. Beside the fact that both positions of that combination are not included in any CDR-L1 canonical template, it is interesting to note that positions L5 and L66 are located at laterally opposite locations with regard to the L1 loop, with a Cα–Cα distance usually around 12 Å and therefore make no contact whatsoever with each other. Keeping all reservations regarding the interpretability of this observation, due to the lack of a dedicated related study on this occasion, this combination appears as a showcase of a possible chained synergy between residues as mentioned earlier. Indeed, as the addition of any Fv position to an already disjoint combination of positions results in a new disjoint combination by definition, then it could be possible that in this case the method captured a chained effect starting at position L5 and involving other or all topological positions between L5 and L66, located on the other side of the L1 loop. Also interestingly, while position L66 is relatively conserved, hosting a Gly in most light Fv sequences, this is apparently not the case in members of the L1-11-II cluster. Instead, the couplet IF fragment sequences L5/L66 were detected to be always different between L1-11-II and all the other clusters in that length. However, only a dedicated study would safely lend itself to such an interpretation, in which case the actual observed combinations of residues of the disjoint signals could play a role in the modelling of the respective cluster’s loop conformation. Finally, important interactions between residues that influence the VH/VL interface orientation can also be potentially captured during DCP, since the defined Interaction Frames involve several residues from both chains.

Important advancements are being made in other methods for the prediction of CDR conformation or loop conformation in general. These include general ab initio modelling techniques (e.g., Loopbuilder, Soto et al., 2008), fragment assembly techniques (e.g., RosettaAntibody, Sivasubramanian et al., 2009; Weitzner et al., 2014), or database search techniques (e.g., FREAD, Choi & Deane, 2011). While the accuracy of these methods is typically measured in average RMSD from the tested crystal structures, respective publications usually do not mention the ratio of wrongly predicted conformations based on an acceptable RMSD threshold, as was the case in this work. To allow future comparisons with such methods, it is worth reporting that this new classification-based prediction method (DCP) presented an average RMSD (and median in Å) after Cα-backbone superposition of CDR residues to the medoids of the correctly identified class of 0.36(0.30)-0.40(0.33)-0.54(0.41)-0.36(0.32) for CDR-L1-L3-H1-H2, respectively (figures calculated from “RMSD distance of observed conformation to cluster medoid”, Supplemental Information 3). For CDR-L1, -L3, -H1 and -H2 with DCP signatures, only 30/692 (∼4%) of correctly identified conformations had an RMSD from the medoid greater than 1 Å (1.03 Å–1.58 Å, average 1.21 Å, median 1.19 Å). A direct comparison between the present implementations of the DCP/canonical templates’ methods and the aforementioned CDR modelling tools was not performed, but can be pursued in the future using the GUI tool mentioned above.

Conclusion

A new predictive model was developed for CDR conformation, its training workflow was designed and a first application was demonstrated on a new test set of structures. Prediction performance was shown to be superior to previous sequence-based methods over all CDRs. The method permits increased parameterisation and presents implementation flexibility. These characteristics allow a considerable margin for performance improvement in future work, and also suggest the possibility that it can be exploited in other fields of biological research. To the best of our knowledge, there existed no similar method with the particular features of DCP at the time of development, i.e., the search for common differences represented by disjoint, mixed sequence combinations between sets of classified sequences, or classified instances in general. Therefore it can be claimed that the method is novel, original and adaptable. It proved impractical to fully verify whether other methods with similar characteristics or features were not developed in research areas other than the biological arena, and therefore this possibility cannot be completely ruled out, e.g., in document-related areas that perform intensive combinatorial operations such as cryptography/decryption. Should this be the case, then only the claim of the method’s novelty regarding the specific application to antibody CDR conformation was demonstrated in this study. In any case and in conclusion, although the development of alternative prediction methods is important, especially ones with an ab initio or fragment-based approach for predicting novel conformations, it is suggested that the strictly sequence-based methods examined here fully retain their innate advantages in prediction time, input simplicity and conformational precision upon positive identification.

Supplemental Information

Supplemental Information 1 Appendix with collection of tables outlining the detected multi-conformation full-rogue clusters

Notable features include resolutions close to 3Å and R-free > 0.25.

Click here for additional data file.

Supplemental Information 2 Appendix: absolute probabilistic significance of IF fragment disjointness

Presentation of a probabilistic closed-form equation for selecting only statistically significant signature signals.

Click here for additional data file.

Supplemental Information 3 Individual predictions per CDR

Detailed tables with all predictions for every CDR in the test sets, along with a measure of RMSD distance of the Query conformation from the closest cluster medoid.

Click here for additional data file.

Supplemental Information 4 Sequence logos for all clusters included in the training set

Sequence logos were constructed for the training clusters using Berkeley’s WebLogo facility (http://weblogo.berkeley.edu/logo.cgi).

Click here for additional data file.

Supplemental Information 5 Detailed canonical templates by CDR/Length

Canonical templates were derived from the clustering set for every applicable conformational cluster, using the definitions of structurally-determining residues described in Martin & Thornton (1996).

Click here for additional data file.

Additional Information and Declarations

Competing Interests

Author Contributions

The authors declare there are no competing interests.

Dimitris Nikoloudis conceived and designed the experiments, performed the experiments, analyzed the data, contributed reagents/materials/analysis tools, wrote the paper, prepared figures and/or tables, reviewed drafts of the paper.

Jim E. Pitts contributed reagents/materials/analysis tools, reviewed drafts of the paper, expert advice, general project supervision.

José W. Saldanha conceived and designed the experiments, analyzed the data, contributed reagents/materials/analysis tools, wrote the paper, reviewed drafts of the paper, expert advice, general project supervision.

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
