# Peer review of "Disjoint combinations profiling (DCP): a new method for the prediction of antibody CDR conformation from sequence"

_PeerJ, doi:10.7717/peerj.455_

## Round 0.1 · original submission · Major Revisions

As mentioned in the previously sent decision for the linked manuscript, I have received comments from two reviewers who read both manuscripts. Both reviewers agree that there is clearly enough substance to each manuscript to stand on itself, but as you will see the current manuscript received more extensive comments that may require you to perform some additional work. Nevertheless, I think that you will be able to accommodate the requests and remarks on this manuscript without much trouble. You can find all the relevant details below.

·

Basic reporting

The manuscript ‘Disjoint combinations profiling (DCP): a new method for the prediction of antibody CDR conformation from sequence’ by Nikoloudis and coworkers describes the results of a machine learning based method (described in a previous manuscript – back-to-back publication with this manuscript); describes the different classes of canonical states of CDRs from Ab.
The manuscript is well written and reads very fluently.

Experimental design

The test and validation set are divided by time. The result of this division is that very closely related Abs might fall within the same set. Often, closely related structures (for example slightly different crystallization conditions, mutants of the Ab, Ag bound or unbound Ab) are submitted at the same moment. I propose to conform the findings also with a different division scheme.
In contrast to the previous manuscript, redundant sequences are removed. It is worth looking at these redundant sequences: do the CDRs have different conformations? And if this is the case, do they fall in different groups / is there a link with Ag binding?

Validity of the findings

In the method presentation section, it is stated that residues originating from the framework also contribute to the CDR conformation. But also the orientation/position of the light chain and heavy chain interface contribute to the conformation of the CDR. This information should be added. Moreover, an analysis of this contribution is desirable.

Additional comments

no comments

Reviewer 2 ·

Basic reporting

This manuscript reported a new method to predict CDR conformations in antibodies using disjoint combinations profiling. In the same journal, the authors also reported the comprehensive classifications of CDRs with all antibody structures available in the PDB by another article accompanying this work. I was requested to peer-review both works.

Although these two works are very related to each other, they are self-contained, so it makes sense to consider publishing both works separately in the same journal.

Experimental design

This work is technically sound.

I have a few suggestions and questions on the experimental procedure.

1. The authors reported the accuracy of this method, and compared it with a canonical template method and H3-rules. I was wondering if the authors could benchmark their method with BLAST to identify correct canonical conformations. A recent work (Weitzner et al. Proteins 2014) showed that BLAST works reasonably well for CDR structure prediction.

2. If I understand, the authors' new method (and a canonical template method) predicts a class of canonical conformations, not the actual coordinates. So, how would the authors build the actual CDR coordinates when they do antibody modeling. On the the hand, BLAST can give us the actual coordinates (templates). In that sense, BLAST may be better choice? Alternatively, it may be better first to predict a class of canonical conformations by the authors' method, then to obtain the templates based on the sequence similarity within a cluster by BLAST?

In the discussion section (page 29), the authors did mention the RMSDs. But I could not follow how they obtain the values by their method. I think the RMSDs reported by the other publications are RMSDs after the refinement process during the modeling. As well known, refinement processes in homology modeling can reduce steric clash and bad geometry, but often deviate template from the "answers". There is a trade-off between reality and accuracy in terms of RMSDs. So, it may not be fair to compare the refined models with raw templates, which could have bad geometries, and to say "our method is better". Also, the way to compute RMSDs need to be clarified. Is the reported RMSD by the authors RMSDs of CDR backbones "after superposing CDRs themselves" or "after superposing frameworks"? There would be only small differences. But it is the small differences that people including the authors are looking at. People who works canonical structures often use the former, but the latter may be more informative for antibody modeler.

3. The authors also reported a set of canonical positions for each CDRs. Does these residue positions match previously reported important positions? Did authors find a new position? Also, I'd like the authors to make sequence logs, so that they can visualize the residue conservations in each position. As the authors clearly stated (page 24), it would be true that statistical consensus sequences may not be the best indicator of canonical clusters, but I think it is still informative.

4. The training set and the test set were separated in this work, which is essential. But it was not clear to me from the manuscript that the training set and the test set have overlap or not. Do they include the "same" antibodies? Also, how do the authors define "non-redundant" structures? These questions may be tricky because the authors analyzed the redundant antibody structures. But it is better that the authors clarify this point.

5. Is the method available in public?

Validity of the findings

The data was interpreted properly. All the associated data was made available on the supplement materials. The original motivation was to improve the prediction of CDRs conformations, and this was achieved by the authors's new method proposed in this manuscript.

In page 8 (last sentence), the authors mentioned that the chained synergies of residue interactions can be captured by the authors' method. Can the authors show such an example?

---

## Round 0.2 · accepted · Accept

Congratulations on the acceptance of your manuscript. I will now also accept your jointly (and related) manuscript.

Reviewer 1 ·

Basic reporting

No comments

Experimental design

No comments.

Validity of the findings

No comments

Additional comments

The revised manuscript ‘Disjoint combinations profiling (DCP): a new method for the prediction of antibody CDR conformation from sequence’ by Nikoloudis and coworkers is now ready for publications. All suggestions were taken into account or addressed. Therefore, I do not have any additional remarks.

Reviewer 2 ·

Basic reporting

All of my previous questions and concerns are addressed by this revision.

Experimental design

No Comments.

Validity of the findings

No Comments.

Additional comments

I really appreciate the authors's efforts in this work. The tools and information provided in the papers are of great help in antibody modeling.